# Ultrasound-Mediated Lysozyme Microbubbles Targeting NOX4 Knockdown Alleviate Cisplatin-Exposed Cochlear Hair Cell Ototoxicity

**DOI:** 10.3390/ijms25137096

**Published:** 2024-06-28

**Authors:** Yuan-Yung Lin, Ai-Ho Liao, Hsiang-Tzu Li, Peng-Yi Jiang, Yi-Chun Lin, Ho-Chiao Chuang, Kuo-Hsing Ma, Hang-Kang Chen, Yi-Tsen Liu, Cheng-Ping Shih, Chih-Hung Wang

**Affiliations:** 1Graduate Institute of Medical Sciences, National Defense Medical Center, No. 161, Sec. 6, Minquan E. Road, Taipei 114201, Taiwan; yking1109@gmail.com (Y.-Y.L.); hwalongchen@yahoo.com.tw (H.-K.C.); 2Department of Otolaryngology—Head and Neck Surgery, Tri-Service General Hospital, National Defense Medical Center, No. 325, Section 2, Cheng-Kung Road, Taipei 11490, Taiwan; lyc_1023@yahoo.com.tw; 3Graduate Institute of Biomedical Engineering, National Taiwan University of Science and Technology, Taipei 106335, Taiwan; aiho@mail.ntust.edu.tw (A.-H.L.); s95027979@gmail.com (H.-T.L.); mlsdmy520@gmail.com (P.-Y.J.); hmps110@gmail.com (Y.-T.L.); 4Department of Biomedical Engineering, National Defense Medical Center, Taipei 114201, Taiwan; 5Department of Mechanical Engineering, National Taipei University of Technology, Taipei 106344, Taiwan; hchuang@mail.ntut.edu.tw; 6Department of Biology and Anatomy, National Defense Medical Center, Taipei 114201, Taiwan; kuohsing91@yahoo.com.tw; 7Division of Otolaryngology, Taipei Veterans General Hospital, Taoyuan Branch, Taoyuan 33052, Taiwan

**Keywords:** ultrasound, microbubbles, cavitation, small interfering RNA (siRNA), cochlear hair cell, cisplatin (CDDP), reactive oxygen species (ROS), NADPH oxidase-4 (NOX4), gene knockdown

## Abstract

The nicotinamide adenine dinucleotide phosphate (NADPH) oxidase 4 (NOX4) protein plays an essential role in the cisplatin (CDDP)-induced generation of reactive oxygen species (ROS). In this study, we evaluated the suitability of ultrasound-mediated lysozyme microbubble (USMB) cavitation to enhance NOX4 siRNA transfection in vitro and ex vivo. Lysozyme-shelled microbubbles (LyzMBs) were constructed and designed for siNOX4 loading as siNOX4/LyzMBs. We investigated different siNOX4-based cell transfection approaches, including naked siNOX4, LyzMB-mixed siNOX4, and siNOX4-loaded LyzMBs, and compared their silencing effects in CDDP-treated HEI-OC1 cells and mouse organ of Corti explants. Transfection efficiencies were evaluated by quantifying the cellular uptake of cyanine 3 (Cy3) fluorescein-labeled siRNA. In vitro experiments showed that the high transfection efficacy (48.18%) of siNOX4 to HEI-OC1 cells mediated by US and siNOX4-loaded LyzMBs significantly inhibited CDDP-induced ROS generation to almost the basal level. The ex vivo CDDP-treated organ of Corti explants of mice showed an even more robust silencing effect of the NOX4 gene in the siNOX4/LyzMB groups treated with US sonication than without US sonication, with a marked abolition of CDDP-induced ROS generation and cytotoxicity. Loading of siNOX4 on LyzMBs can stabilize siNOX4 and prevent its degradation, thereby enhancing the transfection and silencing effects when combined with US sonication. This USMB-derived therapy modality for alleviating CDDP-induced ototoxicity may be suitable for future clinical applications.

## 1. Introduction

The COVID-19 pandemic caused by the SARS-CoV-2 virus resulted in the rapid development of messenger ribonucleic acid (mRNA) vaccines, and this has evolved into the adoption of nucleic-acid-based therapeutics based on antisense oligonucleotides, small interfering RNAs (siRNAs), microRNAs, and mRNAs as tools that further expand treatment options at the genetic level [1,2]. The highly conserved endogenous process of RNA interference (RNAi) exploits small RNAs and negatively regulates gene expression by degradation and/or translational inhibition of target cytoplasmic mRNAs [3,4,5]. RNAi is a remarkable endogenous regulatory pathway and has been investigated for the development of gene-silencing technologies for use in a wide range of applications, ranging from individual gene functions and high-throughput genetic screening to sequence-targeted precision medical therapy and crop improvement [6]. Double-stranded siRNA molecules with sizes of 21–29 nucleotides can intervene in targeted gene expression in the cytoplasm via a specific mechanism of complementary destruction of mRNA, thereby representing a strategy that utilizes an advanced step above the endogenous silencing pathway [7]. This strategy has great potential for the development of drugs designed to knock down the expression of damage- and disease-causing genes [8,9,10].

One particularly attractive target organ for the potential application of this type of gene therapy involving RNAi is the cochlea of the ear. However, this organ has severe restrictions due to the blood–labyrinth barrier; therefore, siRNA has to be delivered to the middle ear by permeation through the round window membrane (RWM) for direct administration into the inner ear [8]. Some previous studies have suggested that efficient delivery of siRNA must take into consideration its small size, negative charge, and high hydrophilicity, as these are features that impair the ability of siRNA to pass through biological membranes [11]. Endosomal entrapment of siRNA is another issue, although several recent chemistry advances and identification of endosomolytic agents, such as polymers, proteins, peptides, and small molecules (e.g., chloroquine), have been successfully employed as siRNA carrier formulations [12]. These advances prompted the present study in which we explore the feasibility of loading siRNA onto lysozyme-shelled microbubbles (LyzMBs), and in combination with ultrasound (US), to determine the targeted knockdown effect in a cochlea model of cisplatin (cis-diaminedichloroplatinum II; CDDP)-induced ototoxicity.

CDDP is a commonly used chemotherapeutic drug for treating numerous malignancies, but its use has a high incidence of ototoxicity (the main adverse effect of this drug) ranging from 20% to 90.1% [13,14]. Several mechanisms underlie CDDP-induced ototoxicity, including increased generation of reactive oxygen species (ROS), DNA damage, activation of apoptosis pathways, and increased calcium influx due to the activation of the transient receptor potential vanilloid 1 channel [15,16]. The transient receptor-potential channels expressed in the inner ear are activated by both capsaicin and CDDP, and they represent essential contributors to CDDP ototoxicity since the knockdown of these channels protects against hearing loss [17].

The cochlea experiences increased oxidative stress mainly in response to the induced expression of several nicotinamide adenine dinucleotide phosphate (NADPH) oxidase isoforms, including NOX1, NOX3, and NOX4, as well as from impairment of the antioxidant defense system [15,16,18]. CDDP treatment promotes NOX4 overexpression in the auditory cells, but this overexpression has been inhibited using NOX4 siRNA to suppress subsequent ROS generation and cytotoxicity [18] Gene therapies for CDDP-induced ototoxicity, in addition to targeting NOX1, NOX3, and NOX4, have also been evaluated by manipulating the expression levels of the other genes, including NTF3, GDNF, HO-1, XIAP, Trpv1, BCL2, Otos, and Nfe2l2 [19]. Previous studies have demonstrated that the diterpenoid trilactone ginkgolide B can protect against CDDP-induced ototoxicity by up-regulating miR214 to suppress the p53-mediated NOX4/p66^shc^ pathway [20,21]. A study by Mei et al. also demonstrated that FER-1 (ferroptosis with ferrostatin-1) can inactivate lipid peroxide radicals and improve mitochondrial function to protect cochlear hair cells from CDDP-induced ototoxicity [22]. These findings confirmed the validity of using siRNA as a therapy; therefore, we explored the possibility of using ultrasound microbubble (USMB)-mediated cavitation for delivery of siRNA to the cochlea.

Microbubbles contain various shell-forming substances, such as albumin, phospholipids, biodegradable polymeric materials, and lysozyme [23,24,25]. Lysozyme microbubbles (LyzMBs) can conjugate with plasmids [26], double-stranded DNA [27], ascorbic acid [28], and polymers [26,29] through electrostatic interactions. The use of vitamin C–loaded LyzMBs combined with US can increase local cochlear concentrations of vitamin C, which then functions to remove ROS and stimulate the production of nitric oxide via endothelial nitric oxide synthase [28]. Gold nanoparticles immobilized on the surface of LyzMBs can significantly improve their antimicrobial efficacy against Micrococcus lysodeikticus [30]. In addition, microbubbles (MBs) are efficient gene carriers and can be used as theranostic tools in imaging-guided gene therapy applications [31]. Stable and relatively monodisperse lysozyme-shelled nanobubbles and LyzMBs can be synthesized using a flow-through sonication technique, with each nanobubble able to carry 1600 double-stranded DNA molecules (oligonucleotides), while protecting the nucleic acids from nucleases [26,27].

This study explored the use of LyzMBs as molecular carriers for cochlear gene therapy in vitro and ex vivo. CDDP promotes NOX4 activation and ROS generation, leading to cochlear damage; therefore, targeting NOX4 knockdown by siRNA and coupling it with enhanced delivery using the USMB cavitation technique may represent a novel therapeutic strategy for alleviating CDDP-induced ototoxicity.

## 2. Results

### 2.1. Ultrasound Microbubble-Mediated siNOX4 Transfection Alleviates CDDP-Induced NOX4 Overexpression and ROS Generation in HEI-OC1 Cells

The effect of CDDP on NOX NADPH oxidase activity in the HEI-OC1 cells was tested by investigating the expression of the isoforms of NOX2, NOX3, and NOX4 genes using qRT-PCR. The ΔCq value of the NOX4 gene was significantly lower in the CDDP-treated group than in the control group (10.05 ± 0.05 vs. 13.71 ± 0.17, *p* < 0.001), indicating that CDDP had caused overexpression of the NOX4 gene in the HEI-OC1 cells. NOX3 expression was not detected in either group. Therefore, we chose NOX4 as the target for gene silencing (Figure 1A).

We also compared different siNOX4-based transfection approaches and their gene-silencing effects by incubating cells with naked siNOX4, a siNOX4 and LyzMBs mixture (siNOX4+LyzMB group), and siNOX4-loaded LyzMBs (siNOX4/LyzMB group) in the CDDP-treated HEI-OC1 cells. Even without US sonication, both siNOX4+LyzMB and siNOX4/LyzMB groups exhibited significant silencing effects compared to the naked siNOX4 group; however, the silencing effect was 17.6% higher in the siNOX4/LyzMB group than in siNOX4+LyzMB group (*p* < 0.05) (Figure 1B). US sonication further enhanced the gene-silencing effects by 39.4% for the siNOX4+LyzMB+US group (*p* < 0.05) and 65.7% for the siNOX4/LyzMB+US group (*p* < 0.01) compared to the respective groups without US. These results suggested that USMBs helped to transfect the siRNA into the cultured HEI-OC1 cells, and that the siRNA, when loaded onto the MBs, underwent a more efficient transfection.

Examination of the changes in CDDP-induced ROS generation revealed an increasing level of ROS in the HEI-OC1 cells after adding CDDP (Figure 1C). All six of the siRNA transfection groups showed significant ROS quenching, but the siNOX4/LyzMB+US group showed the strongest ROS quenching, at about 1.1-fold higher than the basal level of the control group without CDDP administration. Comparison of the NOX4 siRNA treatment groups with and without US sonication showed no significant differences in the ROS levels, indicating that US sonication may not significantly contribute to additional ROS generation.

### 2.2. Characterization of Cy3 siRNA Loading onto LyzMBs

Figure 2A shows that the diameters of the LyzMBs and LyzMBs loaded with Cyanine-3 (Cy3) siRNA (Cy3 siRNA/LyzMBs) were 2.66 ± 0.04 µm and 4.01 ± 0.26 μm, respectively. The zeta potentials of the LyzMBs and Cy3 siRNA dispersed in an aqueous solution were +54.4 ± 1.21 mV and −25.2 ± 5.99 mV, respectively (Figure 2B). The change in the zeta potentials in Cy3 siRNA/LyzMBs (+47.1 ± 0.57 mV) confirmed the electrostatic interaction between the cargo and the carrier. The concentrations of LyzMBs and Cy3 siRNA/LyzMBs were 2.40 ± 0.69 × 10^8^/mL and 1.41 ± 0.07 × 10^8^/mL, respectively (Figure 2C).

Scanning electron microscopy images demonstrated the nanoscale complex of LyzMBs (Figure 3A,B) and Cy3 siRNA loading on the LyzMB shell surface (Figure 3C). At Cy3 siRNA or Cy3 NOX4 siRNA of 300, 400, and 500 ng, the measured adsorption rates for Cy3 siRNA onto LyzMBs (Cy3 siRNA/LyzMBs) were 69.46 ± 7.21% (300 ng), 65.07 ± 3.46% (400 ng), and 65.92 ± 4.50 (500 ng). The corresponding values of Cy3 NOX4 siRNA on LyzMBs (Cy3 siNOX4/LyzMBs) were 89.25 ± 1.11% (300 ng), 86.83 ± 2.25% (400 ng), and 90.16 ± 1.10% (500 ng) (Figure 3F). The adsorption rates for all three siRNA concentrations were all close to the saturation value, and the results obtained for 300 ng were relatively stable during the experiments; therefore, we chose 300 ng as the amount for all subsequent experiments. The calculated concentration of siNOX4 for transfection was 5.6 nM.

### 2.3. Optimizing LyzMB Concentrations for US Sonication Using High-Frequency US Imaging

Using high-frequency US imaging, we investigated the optimal in vitro concentration of LyzMBs for US sonication (Figure 4A). By increasing the number of US sonications from one to six (30 s each time), we found that a LyzMB concentration of 1.2 × 10^7^ MB/mL and two 30 s US sonications at a power density of 1 W/cm^2^ gave an MB destruction efficiency of 59.03 ± 0.53% (Figure 4B), and this was deemed the optimal setting for conducting subsequent experiments. The destruction efficiency at a lower concentration of 6 × 10^6^ MB/mL at the same US power density gave a higher efficiency of 73.34 ± 1.43%; however, the application of USMB tends to cause cells to detach from the well.

### 2.4. CDDP and High Concentrations of LyzMBs Reduce the Viability of HEI-OC1 Cells

Dithiothreitol (DTT) may cause toxicity and impair cell viability. Therefore, in LyzMB preparation, up to five centrifugations were performed to eliminate the influence of DTT on LyzMBs and the cells (Figure 5A). Cell viability decreased as the concentration of LyzMBs increased, and it was also significantly negatively impacted after US sonication. Figure 5B shows that the administration of CDDP alone reduced cell viability to 52.14 ± 0.48%, whereas when the CDDP treatment was followed by LyzMB incubation with or without US sonication, the cell viability decreased further to 20.25 ± 0.23% and 28.26 ± 0.26%, respectively. On the contrary, when cells were treated first with LyzMBs, with or without US, and then with CDDP, cell viability was 61.91 ± 1.27% and 67.68 ± 0.67%, respectively. Cell viabilities of 20.25 ± 0.23% and 28.26 ± 0.26% were too low to perform subsequent gene knockdown experiments; therefore, we chose to treat the cells with LyzMBs before the CDDP treatment in all subsequent experiments.

### 2.5. US-Mediated Cy3 siRNA/LyzMB Cavitation Enhances In Vitro siRNA Transfection Efficiency

Fluorescence microscopy images indicated the presence of Cy3 siRNA (labeled in red) in all Cy3 siRNA-based transfection groups, as shown by Cy3 siRNA fragments in the perinuclear area (Figure 6A,B). When combined with US sonication, treatment with LyzMBs, either mixed or loaded with Cy3 siRNA, exhibited more robust fluorescence. Comparison of the changes in transfection efficiency after US sonication revealed transfection efficiencies of 22.06 ± 2.48% for the Cy3 siRNA+LyzMB group without US vs. 33.64 ± 1.44% with US (*p* = 0.002), whereas the efficiencies for the Cy3 siRNA/LyzMB group were 19.78 ± 2.11% without US and 48.18 ± 2.82% with US (*p* < 0.001). These data suggested that USMB cavitation enhanced siRNA transfection into the cells, as transfection was more effective when the siRNA was loaded onto LyzMBs than when simply mixed with the LyzMBs (Figure 6C).

### 2.6. US-Mediated siNOX4/LyzMB Transfection Significantly Increased the Efficiency of NOX4 siRNA Transfection in CDDP-Treated Auditory Cells

We also tested the Cy3 NOX4 siRNA transfection efficiency either mixed with LyzMBs (Cy3 siNOX4+LyzMB group) or loaded onto LyzMBs (Cy3 siNOX4/LyzMBs group) with and without US sonication (Figure 7A,B). Again, transfection was higher when siRNA was loaded onto the LyzMBs than when simply mixed with the LyzMBs (24.90 ± 2.32% vs. 18.68 ± 2.08%, *p* = 0.0298) and was further improved using US sonication (Figure 7C). The use of naked Cy3 siNOX4 did not result in significant siRNA delivery.

Interestingly, transfection with siNOX4/LyzMBs with US (52.79 ± 2.50%) or without US (21.04 ± 0.68%) was more effective when performed before the CDDP treatment (Appendix A). These data suggest that a prior siNOX4 transfection using siNOX4/LyzMB can increase the survival of CDDP-treated cells.

### 2.7. Immunohistochemistry Demonstrates USMB-Mediated NOX4 siRNA Transfection Significantly Attenuates Ex Vivo CDDP-Induced Cochlear NOX4 Expression and Ototoxicity

The immunostaining results shown in Figure 8A confirm that NOX4 was highly expressed in the organ of Corti, including the inner and outer hair cells, in the CDDP control group. All NOX4 siRNA-transfected groups showed varying degrees of NOX4 silencing effects. Among the transfected groups, the group that combined siNOX4 transfection and USMB delivery (siNOX4/LyzMBs+US group) showed significantly more efficient transfection than the other groups (*p* < 0.001) (Figure 8B).

Examination of the organ of Corti revealed significant morphological damage such as disorganized arrangement of outer hair cells with destruction of sensory epithelium and disruption and loss of stereociliary bundles of outer hair cells (Figure 9), suggesting severe CDDP-mediated ototoxicity. Pre-transfected with different NOX4 siRNAs, preservation of relatively intact sensory epithelium and outer hair cell bundles can be observed, most pronounced in the siNOX4/LyzMBs combined with the US group, followed by siNOX4/LyzMBs and then naked siNOX4 groups.

## 3. Discussion

This study demonstrated that better NOX4 knockdown and better protection of cochlear hair cells against CDDP cytotoxicity are achieved with US-mediated siNOX4/LyzMBs than with NOX4 siRNA alone. This is the first study to assess the feasibility of using USMBs to enhance siRNA transfection for targeted gene silencing. USMB-mediated cavitation increases cell membrane permeability; therefore, we saw improved NOX4 siRNA transfection to the auditory cells and suppression of CDDP-induced NOX4 gene expression and subsequent ROS generation.

The selection of NOX4 as a candidate gene suitable for siRNA silencing in HEI-OC1 cells treated with CDDP was based on the significant decrease in the ΔCq value of the NOX4 gene, as this indicated that CDDP induced an overexpression of NOX4 and that NOX4 should be highly susceptible to siRNA-mediated gene silencing. In humans, NOX4 is most abundantly expressed in the kidneys and lungs, but the signaling pathways associated with NOX4 are complicated. Both negative and positive feedback are influential in regulating NOX4 expression in many disorders, including pulmonary diseases, kidney disease, and cancers [32,33,34,35]. Kim et al. demonstrated that activation of NADPH oxidase contributed to CDDP ototoxicity [18,36], in agreement with our findings, suggesting that ROS generated through the activation of NOX4 may play an essential role in CDDP ototoxicity.

CDDP is presently the standard chemotherapeutic agent used to treat squamous cell carcinoma of the head and neck [37,38]; however, it manifests adverse effects in the form of bilateral and irreversible sensorineural hearing loss that negatively impacts the quality of life, especially in the pediatric population [39]. Although the induction of specific isoforms of NOX, including NOX1, NOX3, and NOX4, has been attributed to CDDP and induced the overproduction of ROS and subsequent ototoxic damage [15,16,18], the present study, provides evidence that targeting NOX4 for silencing diminishes the level of ROS as part of its therapeutic effect.

The findings presented here also showed that the use of USMBs can also improve transfection efficiencies. USMBs can initiate both stable and inertial cavitation effects, in which stable cavitation produces microstreaming around the MBs, whereas inertial cavitation creates a shock wave accompanied by sonic-speed microinjection at the cell surface [40]. In the field of medical ultrasound, the peak acoustic negative pressure (also known as the peak rarefaction pressure) is a critical parameter for the occurrence of cavitation, particularly in pre-existing gas bubbles in water [41,42,43,44]. Short pulse ultrasound can induce inertial cavitation between 0.5–2 MPa peak negative pressure at 1 MHz [45]. However, due to variations in the size and shape of individual bubbles, the strength of the shock front, and the positive pressure amplitude, the peak negative pressure of the incident wave alone cannot be used as a sole indicator of the threshold [46]. Acoustic emissions can be captured and analyzed using a wideband polyvinylidene fluoride hydrophone for spectral content and pressure matching [47,48]. In this study, the acoustic power intensity was measured using the Radiation Force Balance [49] (model UPM-DT-1, Ohmic Instruments, Easton, MD, USA). According to our experimental setup (all specimens placed 5 mm in front of the transducer), the acoustic power was determined to be 240 mW at a duty cycle of 50% throughout the experiments. This intensity can be estimated to be equivalent to a spatial peak temporal average intensity acoustic intensity (*I*_SPTA_) of 213 mW/cm^2^.

Comparison between the LyzMBs merely mixed with NOX4 siRNA and the LyzMBs with NOX4 siRNA loading onto their surfaces revealed a much better transfection with the loaded forms. LyzMBs have a positive charge, so the surface potential exceeds zero, meaning that they can attract molecules that have negative charges. For this reason, the siRNA shell readily adsorbs onto LyzMBs by electrical adsorption because the presence of phosphate groups in siRNA creates a global negative electrical charge [50], resulting in the formation of a diffuse siRNA coating on the LyzMBs to generate siRNA/LyzMBs. These differences may explain why NOX4 siRNA loading onto LyzMBs can enable a higher transfection efficiency than is achieved by mixing siRNA with LyzMBs.

In general, siRNA transfections are suggested to use siRNA concentrations of 5–100 nM [51,52,53,54]. Theoretically, a higher siRNA concentration is expected to have a greater expected mRNA silencing effect. However, the optimal effective siRNA concentration depends on the target and the cell type and can also be dictated by unwanted off-target effects, which can occur at high siRNA concentrations. These can include activation of the interferon response and unintended triggering of genes that have only a low sequence homology to the RNA molecule [53,55]. In this study, we used various siRNAs, including NOX4 siRNA, Cy3 siRNA, and Cy3 siNOX4, at the minimal concentration of 5 nM, which helped to provide an accessible quantification of RNA interference and to differentiate the novel benefits afforded by the use of USMBs.

Preparation of the siCy3-loaded or siNOX4-loaded LyzMB relied on the ability of these composite molecules to undergo spontaneous self-assembly through electrostatic interactions between the anionic siRNA and a cationic LyzMB, similar to the dendriplexes that use cationic dendrimers to carry anionic siRNA [56]. Previously, we demonstrated the USMB technique as a drug delivery system that enables the specific transport of a drug into particular cell types, tissues, and tumors while reducing undesired systemic side effects [40,57,58,59]. The results shown in the present study support the future application of USMBs as siRNA carriers that enhance siRNA transfection efficiency after topical US sonication.

## 4. Materials and Methods

### 4.1. Preparation of LyzMB-Loaded siRNA

Figure 10 shows the self-assembly process used to produce siRNA-coating LyzMBs. The LyzMBs were first prepared according to our previously described procedure [23,60]. In brief, 50 mg of chicken–egg-white Lyz was dissolved in 1 mL of 1 M Tris buffer (pH 8.0), 20 mg of reducing agent DL-dithiothreitol (DTT) was then added, and the solution was shaken at 50 rpm for 15 min at 4 °C to allow sufficient time for partial reduction to occur. The LyzMBs were generated by sonicating this solution in perfluoropropane gas (C3F8) using a sonicator at a power of 120 W (Branson Ultrasonics, Danbury, CT, USA) for 30 s. The LyzMBs were centrifuged at 1200 rpm (110× *g*; F2402 rotor, Beckman Coulter, Fullerton, CA, USA) for 2 min and then washed five times to eliminate the Tris buffer and DTT using Milli-Q water (pH 6.4, resistivity = 18.2 mW at 25 °C).

SiGENOME SMARTpool NOX4 siRNA was purchased from Dharmacon. Cy3-labeled siRNA (Cy3 siRNA) and Cy3-labeled NOX4 siRNA (Cy3-siNOX4) were obtained from BIOTOOLS, Taipei, Taiwan. Various siRNA-loaded LyzMBs were prepared by incubating the original Cy3 siRNA and Cy3-siNOX4 at 300 ng/mL, 400 ng/mL, or 500 ng/mL with the produced LyzMBs on a rotary shaker (50 rpm; Shaker RS-01, TKS, New Taipei City, Taiwan) for 30 min at 4 °C in a refrigerator to produce Cy3 siRNA/LyzMBs and Cy3-siNOX4/LyzMBs, respectively. These siRNA-loaded LyzMBs were washed once to remove unbound siRNA. An ELISA reader (Epoch, Biotek, Winooski, VT, USA) was then used at excitation and emission wavelengths of 500 nm and 600 nm, respectively, to calculate the adsorption efficiency of the Cy3 siRNA and Cy3-siNOX4 onto the LyzMBs, with the results substituted into the following equation:(1)$$Adsorption\;Efficiency\;(\%)=\frac{Adsorption\;capacity\;(ng)}{Total\;siRNA\;(ng)}\times 100\%$$

The numbers of LyzMBs and siRNA/LyzMBs in the solution were measured using a MultiSizer III device (Beckman Coulter) with a 30 mm aperture probe whose measurement boundary ranged from 0.6 mm to 20 mm. The suspension size distribution and zeta potential were measured using dynamic light scattering (Nanoparticle Analyzer, Horiba, Kyoto, Japan). The morphology of the siRNA/LyzMBs was examined by filtering 40-fold-diluted siRNA/LyzMBs through a 5 mm syringe filter (Sartorius, Goettingen, Germany), and then analyzing 5 mL of the filtered siRNA/LyzMBs by scanning electron microscopy (SEM) after coating the samples with platinum at 20 mA for 20 min using an automatic sputter coater (JFC-1300, JEOL, Tokyo, Japan). The SEM images were obtained at an accelerating voltage of 15 kV.

### 4.2. Optimization of LyzMB Concentrations and US Parameters for the Destruction of siRNA-Loaded LyzMBs

Since the enhancement of drug delivery is related to the destruction efficacy of MBs [61], we investigated the US parameters required for LyzMB destruction in vitro by subjecting the MBs to US at 1 W/cm^2^ (*I*_SPTA_ = 213 mW/cm^2^; ST2000V, Nepa Gene, Ichikawa, Japan) for 30 s, with 1 to 6 replications of the US treatment. We set the US device (Nepagene) equipped with a 10 mm diameter probe to operate at a center frequency of 1 MHz and a duty cycle of 50%. The probe was placed directly onto the cover of a 24-well plate using gel as a coupling agent. Each well of the 24-well plate was filled with 4 mL of LyzMBs at either 2.4 × 10^7^ MB/mL, 1.2 × 10^7^ MB/mL, or 0.6 × 10^7^ MB/mL. After completing the US sonication, the LyzMB solution in each well was diluted tenfold and then imaged using a US animal imaging system (Prospect, S-Sharp Corporation, Taipei, Taiwan). All images were processed using custom MATLAB programs (The MathWorks, Natick, MA, USA), and the destruction efficiency was evaluated by calculating the difference in the gradient strength on the LyzMB images before and after US sonication. The in vitro effects of US-mediated LyzMB destruction on HEI-OC1 cells were evaluated by placing 3 × 10^4^ cells in each well of a 24-well plate and incubating overnight. The next day, each well was filled with 4 mL of MBs at the desired concentration, determined based on the previous experimental results, followed by US sonication. After the US sonication, the LyzMB solution was replaced with a culture medium (DMEM, Invitrogen, Waltham, MA, USA, without FBS), and the cells were allowed to grow for 24 h.

### 4.3. Cell Culture and CDDP Treatment

The auditory HEI-OC1 cell line was kindly provided by Dr. Federico Kalinec (House Ear Institute, Los Angeles, CA, USA) and maintained in high-glucose DMEM (Invitrogen) containing 10% FBS at 33 °C and 5% CO_2_. For CDDP treatment, cells were seeded at a density of 3 × 10^4^ cells/well per 24-well plate for 24 h, followed by incubation with 20 μM CDDP for 24 h. The cell viability assay was conducted by adding 500 µL of Alamar Blue (1%) to each well for 1 h, and then the optical density (OD) of each culture well was measured with a microplate reader (Epoch^TM^) at excitation and emission wavelengths of 560 nm and 590 nm, respectively. The OD in the control cell group was taken to indicate a viability of 100%.

### 4.4. CDDP Administration and NOX4 siRNA Transfection In Vitro

HEI-OC1 cells at a density of 3 × 10^4^ cells/well per 24-well plate were pretreated with 20 μM CDDP for 24 h, then naked siNOX4, a siNOX4 and LyzMBs mixture (siNOX4+LyzMBs group), or siNOX4-loaded LyzMBs (siNOX4/LyzMBs group) were added with the TOOLSmartFect transfection reagent (TOOLS) for transfection for 1.5 h. The treatments included groups of cells treated with or without US sonication. Another set of HEI-OC1 cells was transfected with the above various siNOX4s. Again, the treatments included groups treated with and without US sonication. After the treatments, all cells were plated and incubated for 4 h, followed by a 24 h incubation with 20 μM CDDP.

### 4.5. Immunocytochemistry and In Vitro siRNA Transfection Efficiency

For immunocytochemistry studies, cells were washed three times with PBS and fixed with 200 µL of 4% paraformaldehyde (Sigma-Aldrich, Saint Louis, MO, USA) in PBS for 5 min at room temperature. After washing three times with PBS, the cells were incubated with 5 μL of DAPI (Southern Biotech, Birmingham, AL, USA) at room temperature for 10 min. Fluorescence immunostaining images of the cells were observed and captured using an inverted fluorescence microscope (CKX-41, Olympus, Tokyo, Japan). Transfection efficiency was quantified by counting the average number of transfected cells showing Cy3 fluorescence relative to the total number of cells showing DAPI nuclear staining in each sampling field.

### 4.6. RNA Isolation and qRT-PCR

Total RNA was extracted from the HEI-OC1 cells treated with CDDP or transfected with various siRNAs using the EasyPrep Cell/Bacteria RNAprep Purification Kit (TOOLS). The samples were reverse transcribed to cDNA using the ToolsQuant II Fast RT Kit (TOOLS) and the TOOLS Easy 2 × Probe qPCR Mix Kit (TOOLS) by quantitative real-time PCR (qPCR) on a real-time PCR system (LightCycler 480 II, Roche, Mannheim, Germany). The primers included NOX4 (Mm00479246_m1) and GAPDH (Mm99999915_g1). The master reaction mixture consisted of 10 mL of 1X TOOLS 2xSYBR qPCR Mix (TOOLS), 2 μL of cDNA template sample, 1 μL of TaqMan gene expression assays (Thermo Fisher Scientific, Waltham, MA, USA), and RNase-free ddH2O was added up to 20 mL. The PCR conditions were as follows: at 37 °C for 2 min, 95 °C for 5 min, 45 cycles of 95 °C for 10 s, and 60 °C for 30 s. Threshold cycle data were analyzed using the analysis of the quantification cycle (Cq) values and the LightCycler^®^ 480 Gene Scanning software v1.5.0. Relative gene expression levels were normalized to the internal control (GAPDH) and expressed as ΔCq values. The changes in the target gene expression relative to those of controls were analyzed using the ΔΔCq method.

### 4.7. ROS Measurement

HEI-OC1 cells at a density of 3 × 10^4^ cells/well per 96-well plate were treated with various siRNA transfection and CDDP treatment, with or without US sonication, and then incubated with 10 μM H_2_DCFDA (Thermo Fisher Scientific) for 1 h at 37 °C in the dark. ROS induced the cleavage of the acetate groups by intracellular esterases and oxidation, thereby converting the nonfluorescent H_2_DCFDA to the highly fluorescent 2′,7′-dichlorofluorescein (DCF). After washing the cells, the DCF fluorescence was measured using a fluorescence plate reader (Synergy H4 Hybrid Reader; BioTek Instruments, Winooski, VT, USA) with excitation at 485 nm and emission at 538 nm.

### 4.8. Cochlear Explant Culture

The Institutional Animal Care and Use Committee of the National Defense Medical Center, Taipei, Taiwan, approved the experimental protocols (approval number: IACUC-20-175). Neonatal mice of the CBA/CaJ strain at postnatal day 3 (P3) were euthanized, and the cochlea from each mouse was removed from the temporal bone. Under a dissection microscope, the organ of Corti was carefully separated from the spiral lamina and spiral ligament and then attached to a glass-bottomed dish (ibidi, Grafelfing, Germany) coated with Cell-Tak (#354240, Corning, Fisher Scientific, Waltham, MA, USA). Warmed culture medium (98% DMEM and 1% N_2_; #17502-048, Thermo Fisher Scientific) and 1% ampicillin (#11593-027, Invitrogen) were added onto the explanted organs of Corti. The explants were incubated (37 °C and 5% CO_2_) for 10–16 h, and then 200 µL of the final culture medium (97% DMEM, 1% FBS, and 1% N_2_ supplement (Thermo Fisher Scientific; 1% ampicillin) was added to submerge the explants. Three different transfections (naked NOX4 siRNA, siNOX4/LyzMBs, or siNOX4/LyzMBs+US) were then administered to the explants for 4 h. The explants were incubated with 20 μM CDDP for a further 24 h.

### 4.9. NOX4 Immunohistochemistry in Explants

The cochlear explants were fixed using 4% paraformaldehyde for 30 min at room temperature and then incubated with anti-NOX4 polyclonal antibodies (1:100; Santa Cruz Biotechnology, Dallas, TX, USA) and anti-myosin-7a polyclonal antibodies (1:100; Novus Biologicals, Centennial, CO, USA) for 2 h at room temperature. After rinsing with PBS, the samples were incubated with Alexa-Fluor-488-conjugated donkey anti-sheep antibodies (1:500; Thermo Fisher Scientific) and Alexa Fluor 555-conjugated goat antirabbit antibodies (1:500; Thermo Fisher Scientific) for 1 h, followed by Alexa-Fluor-647-conjugated phalloidin incubation for 30 min, mounted in DAPI (4′,6-diamidino-2-phenylindole) Fluoromount-G mounting medium (Southern Biotech), and covered with a coverslip. Images were acquired using a confocal laser scanning microscope (Zeiss LSM 880, Carl Zeiss, Jena, Germany). The immunostaining intensity of NOX4 in the images was quantified using ImageJ software bundled with 64-bit Java 1.8.0_172 (https://imagej.nih.gov/ij/download) (accessed on 20 June 2024). The staining intensities were expressed in arbitrary units and subjected to histogram analysis.

### 4.10. Statistical Analysis

The data were analyzed statistically using the two-tailed Student’s *t*-test to compare two groups. Multiple groups were compared using one-way ANOVA, followed by the Bonferroni multiple-comparisons test. A probability value of *p* < 0.05 was considered statistically significant.

## 5. Conclusions

This study first established a platform for using siNOX4-loaded LyzMBs combined with US to diminish CDDP-induced ROS generation and subsequent ototoxicity. The USMB-mediated cavitation increases the permeability of the transfected cells, while the cationic LyzMBs promote the attraction of anionic siNOX4 for loading. Once loaded, the siNOX4 is protected from degradation. The USMB technique overcomes the electrostatic repulsion of the cell membrane to effectively deliver siNOX4 into cultured auditory cells and the organ of Corti. Most importantly, USMB-assisted siRNA transfection can be applied to specific organs and tissues with minimal siRNA concentrations, thereby decreasing or avoiding undesired side effects by administering systemic or high siRNA concentrations.

## Figures and Tables

**Figure 1 ijms-25-07096-f001:**
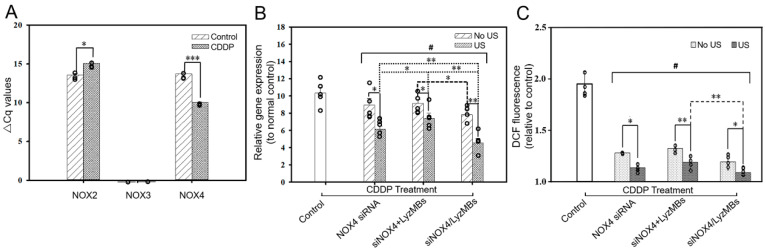
(**A**) Quantitative real-time reverse-transcription PCR-based detection of the mRNA expression levels of various NOX genes in CDDP-treated HEI-OC1 cells. (**B**) Silencing of the NOX4 gene in CDDP-treated HEI-OC1 cells using different siNOX4-based approaches, with or without US. (**C**) ROS quenching in CDDP-treated HEI-OC1 cells using different siNOX4-based approaches with or without US. * *p* < 0.05, ** *p* < 0.01, *** *p* < 0.001; # (One-way ANOVA with Bonferroni multiple comparisons test; *p* < 0.001). Values are expressed as the mean ± standard error of the mean (SEM), with *n* = 5 for each bar.

**Figure 2 ijms-25-07096-f002:**
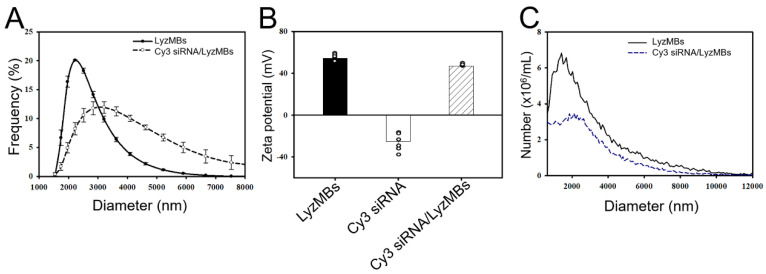
Frequency-dependent size distributions (**A**), zeta potentials (**B**), and concentrations (**C**) of LyzMBs and of Cy3 siRNA-loaded LyzMBs. Values are expressed as mean ± SEM (*n* = 5).

**Figure 3 ijms-25-07096-f003:**
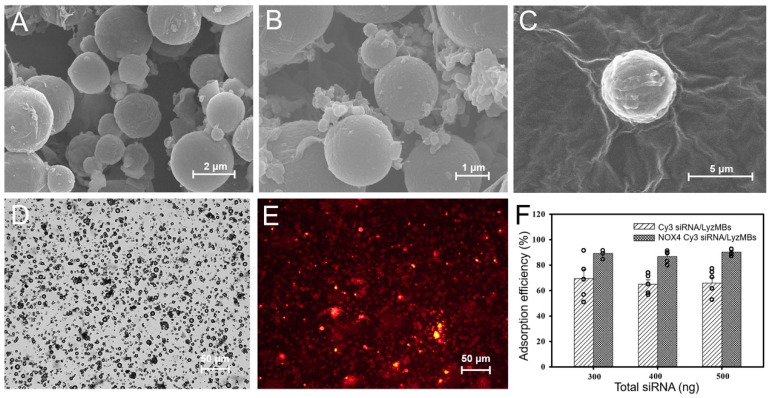
Scanning electron microscopy images of LyzMBs (**A**), images at higher magnification (**B**), and a single Cy3 siRNA-loaded LyzMB (**C**). The bright-field image of LyzMBs (**D**). Fluorescence microscopy image of Cy3 siRNA-loaded LyzMBs (red fluorescence) (**E**). Adsorption efficiency of Cy3 siRNA and Cy3 NOX4 siRNA onto the LyzMBs at different initial siRNA concentrations (**F**). Data are expressed as mean ± SEM, with *n* = 5 for each bar.

**Figure 4 ijms-25-07096-f004:**
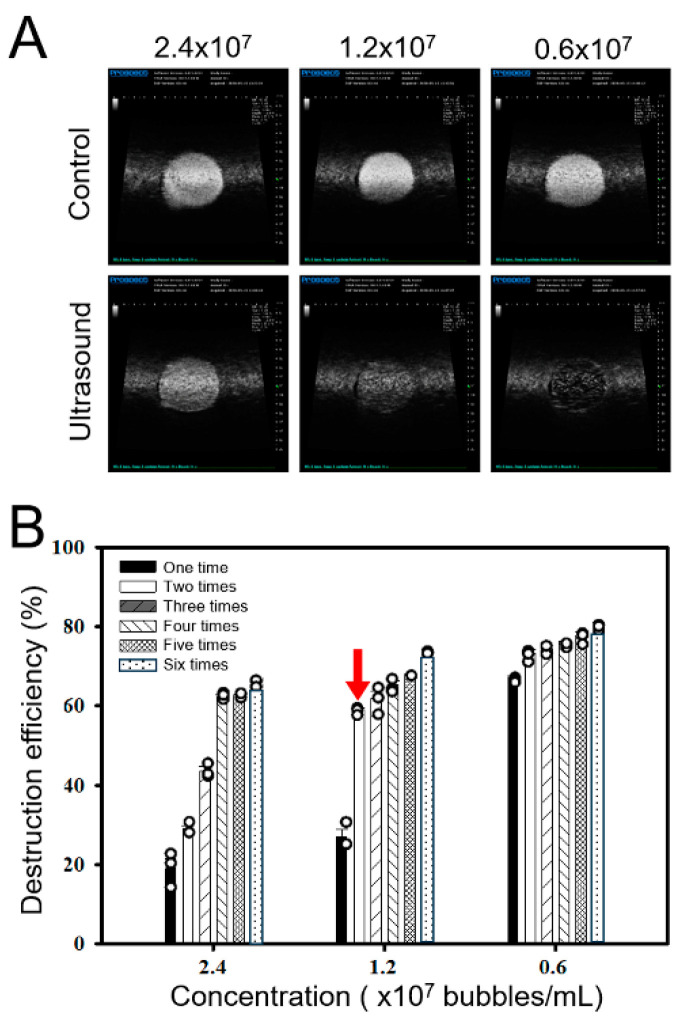
High-frequency US images of MBs without (Control) and with US sonication. (**A**) Images of MBs at different concentrations after two US sonications for 30 s. (**B**) Quantification of MB destruction for MBs at different concentrations with US sonication for 30 s one to six times. The red arrow indicates the optimal MB concentration and US parameter used in the subsequent in vitro and ex vivo experiments. Data are expressed as mean ± SEM, with *n* = 5 for each bar.

**Figure 5 ijms-25-07096-f005:**
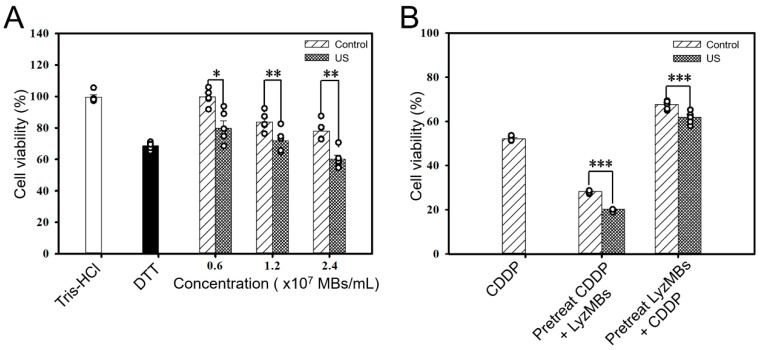
(**A**) Quantitative results for the cell viability effects of Tris-HCl, DTT, and LyzMBs + ultrasound (US) at different concentrations. (**B**) Quantitative results for cell viability of CDDP treatment alone, CDDP pretreatment followed by LyzMBs with or without US, and LyzMBs pretreatment with or without US followed by CDDP administration. * *p* < 0.05, ** *p* < 0.01, *** *p* < 0.001. Data are expressed as mean ± SEM, with *n* = 5 for each bar.

**Figure 6 ijms-25-07096-f006:**
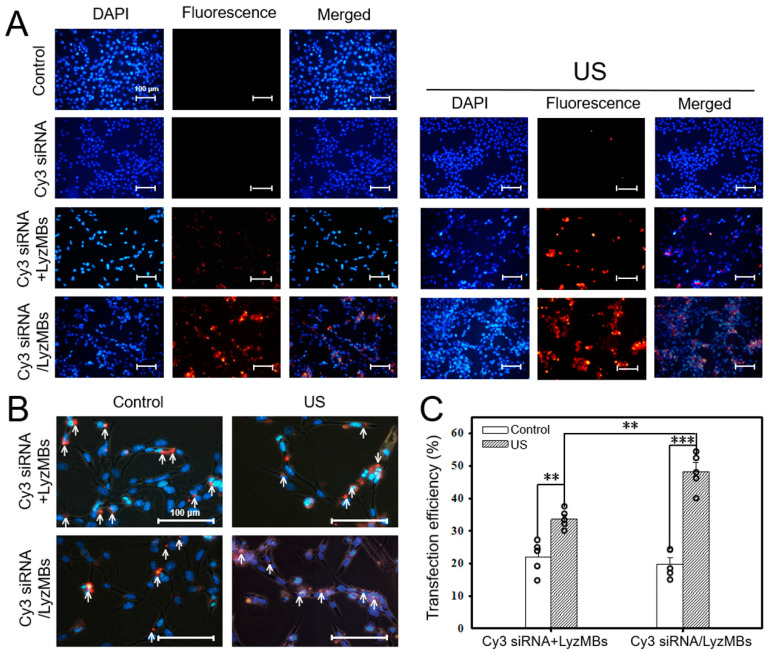
Fluorescence microscopy images (**A**) and at higher magnification (**B**) showing the Cy3 siRNA transfection efficiencies of HEI-OC1 cells in various treatment groups. (Cy3 siRNA+LyzMBs = Cy3 siRNA mixed with LyzMBs; Cy3 siRNA/LyzMBs = Cy3 siRNA loaded onto LyzMBs). Arrows indicate successfully transfected cells containing Cy3 siRNA (red fluorescence); DAPI-stained nuclei (blue fluorescence). Quantification of the Cy3 siRNA transfection efficiencies with or without US sonication (**C**). Scale bar = 100 μm. ** *p* < 0.01, *** *p* < 0.001. Data are expressed as mean ± SEM, with *n* = 5 for each bar.

**Figure 7 ijms-25-07096-f007:**
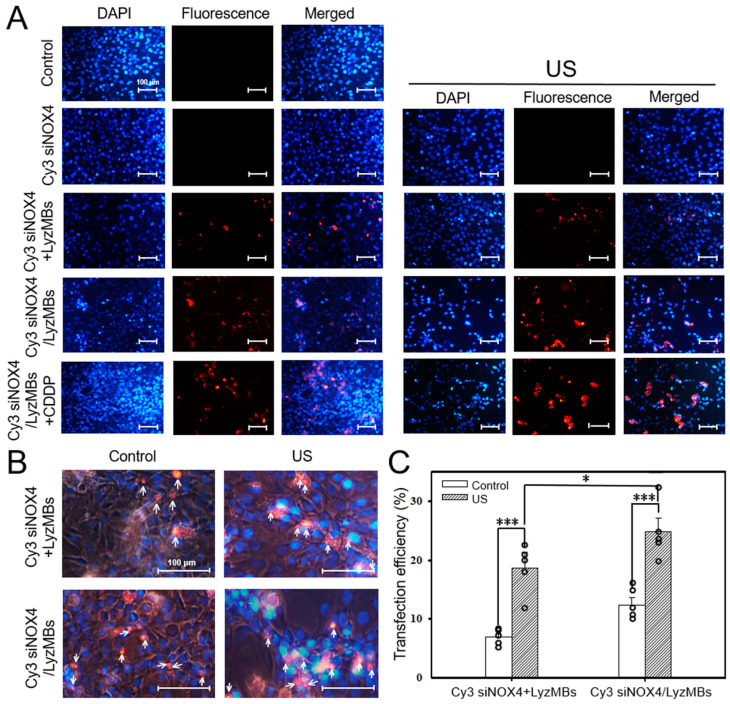
Fluorescence microscopy images (**A**) and at higher magnification (**B**) show the Cy3 siNOX4 transfection efficiencies for HEI-OC1 cells in the various treatment groups. Arrows indicate the presence of siNOX4 (red fluorescence); DAPI-stained nuclei (blue fluorescence). Quantification of the siNOX4 transfection efficiencies for HEI-OC1 cells (**C**). Scale bar = 100 μm. * *p* < 0.05, *** *p* < 0.001. Data are expressed as mean ± SEM, with *n* = 5 for each bar.

**Figure 8 ijms-25-07096-f008:**
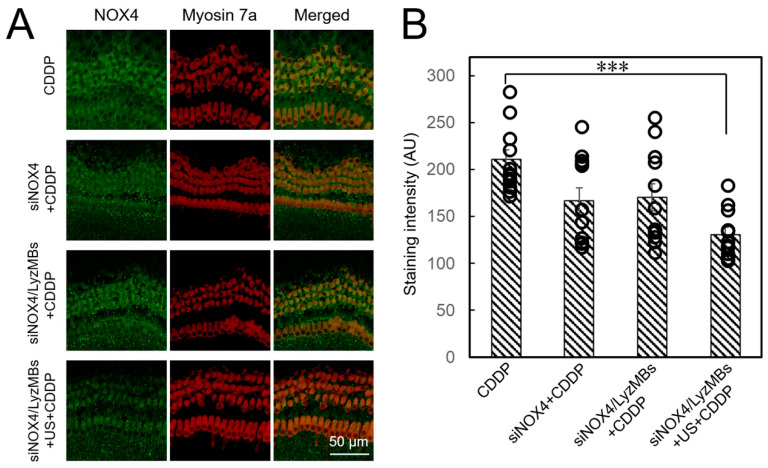
(**A**) Immunostaining of NOX4 in explant cultures of the organ of Corti from neonatal mice pretreated with NOX4 siRNA alone, siNOX4/LyzMBs, or siNOX4/LyzMBs combined with ultrasound sonication, followed by CDDP treatment. (**B**) Histogram of the fluorescence intensities of NOX4 in the organ of Corti. Stained with antibodies to NOX4 (green) and myosin 7a (red). Scale bar = 50 μm. Data are expressed as mean ± SEM, with *n* = 12 for each bar. *** *p* < 0.001.

**Figure 9 ijms-25-07096-f009:**
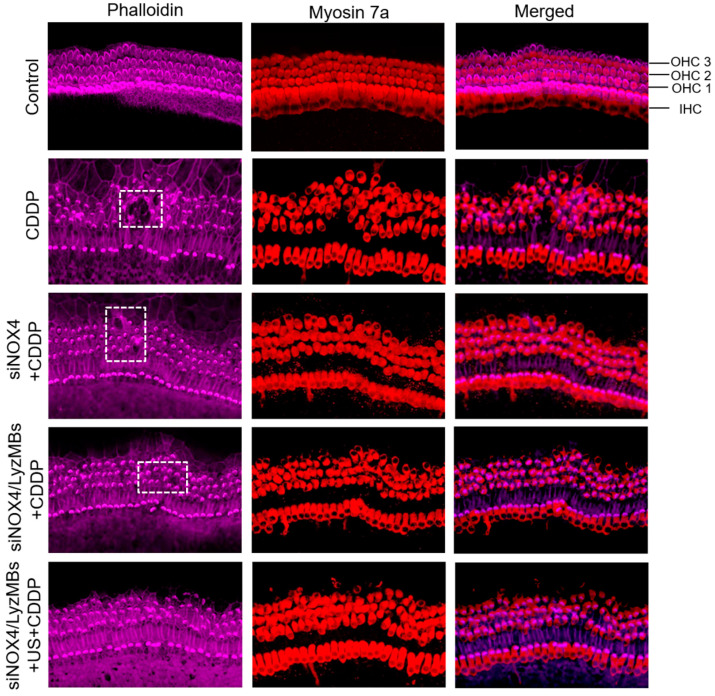
Representative images of CDDP-induced hair cell damage in organ of Corti explant cultures of the middle turn pre-transfected with NOX4 siRNA alone or with siNOX4-loaded LyzMBs without or with ultrasound sonication prior to CDDP treatment. Dashed squares indicate the destruction of sensory epithelium. Stained with antibodies to phalloidin (purple) and myosin 7a (red). Scale bar = 50 μm. OHC, outer hair cells; IHC, inner hair cells.

**Figure 10 ijms-25-07096-f010:**
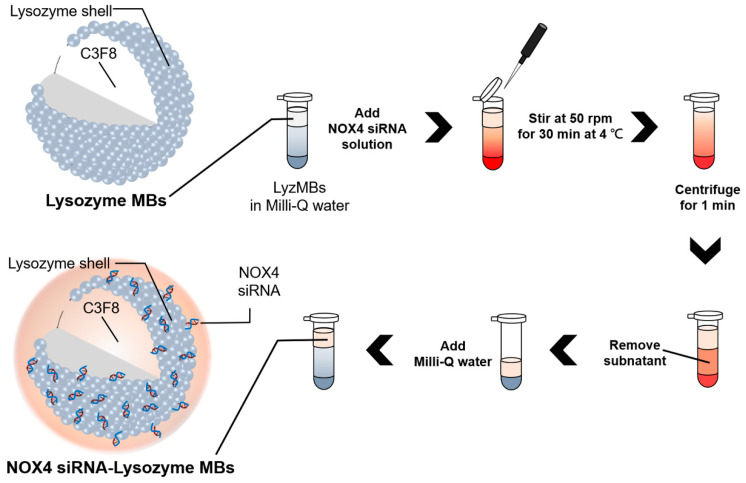
Schematic (not to scale) of the self-assembly process used to produce a negatively charged NOX4 siRNA coating by electrical adsorption onto positively charged lysozyme-shelled MBs.

## Data Availability

The data presented in this study are available upon reasonable request from the corresponding author.

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
