# Peer review of "Ultrasound-Mediated Lysozyme Microbubbles Targeting NOX4 Knockdown Alleviate Cisplatin-Exposed Cochlear Hair Cell Ototoxicity"

_ijms, 2024, doi:10.3390/ijms25137096_

Round 1

Reviewer 1 Report

Comments and Suggestions for Authors

The authors present an experimental study aimed at reducing chemotherapy-induced ototoxicity. The paper is well organized and thorough. Some further details are needed, as listed below. 

L102: There is gap between establishing the validity of siRNA therapy for ototoxicity and using a USMB cavitation procedure. The authors need to clarify the existing deficiency that they propose to fix. In other words, why is the USMB + siRNA technique hypothesised to be better?

L160: 'indicating that US sonication did not stimulate ROS generation.' This is not directly supported by the data - consider restating.

L191-198: Perhaps some symbols or other text are missing. See for example L193 shows as 'concentration of 1.2    107 MB/ML...'

L197: '... too many cultured cells were detached...', define what constituted 'too many'?

L211: Do the authors have confidence in their viability quantification to a second decimal place? 

L213: What was the time separation between LyzMB and CDDP treatments?

L234-236: This is a very important finding, since very often bubbles and drugs are co-administered for simplicity. Can the authors discuss the mechanistic difference(s) that might explain this finding. 

Figure 9: It would greatly help the reader's appreciation of the therapy to have images of healthy controls for comparison.

L316: Bubbles were destroyed during ultrasound exposure, but this does not by itself require inertial cavitation. For example, there may be some cracking of the shell under ultrasound loading. Given this, and the fact that the authors didn't directly monitor bubble dynamics during cell/organ exposures, it would be appropriate to remove the 'inertial cavitation' comment.

L346: Generally well-described materials/methods. 

L348: Of the many possible carrier constructs, why were lysozymes chosen for this application?

L374: Units are probably micrometers, not millimeters

L388: How were the initial ultrasound parameters chosen? Was there a prior optimization pilot study? If not, isn't it possible that other parameters (pressure, pulse length, duty cycle, frequency) could have worked even better?

L397: Was the time stability of the LyzMB ever assessed?

L403: For bubble destruction, rarefactional pressure is a key parameter. The use of a device rated in terms of power instead of pressure, along with testing through/in a well plate where the fields may be complex, all limits the interpretation and translatability of the technique beyond the experiments presented here. The authors should at least acknowledge this issue in the discussion section. 

L418: Would the use of a transfection reagent be necessary for the clinical application, and if so, what constraints does this place on the proposed technique?  

L499: Can the authors comment on how the proposed technique would work clinically? Specifically, how might the treatment be administered to reach the cochlea and expose to suitable ultrasound levels in a targeted manner?

Comments on the Quality of English Language

minor editing only

Reviewer 2 Report

Comments and Suggestions for Authors

FIGURE 9:  What areas of the cochlea are displayed? What is the morphology of the rest of the cochlea?All areas of the cochlea should be displayed.

Also, it is difficult to discern any differences in hair cell preservation among the three NOX 4 siRNA plus cisplatin treated groups. Please clarify.

Discussion The above concern is crucial to determine the validity of the conclusions expressed in the Discussion section.

How would the microbubble-siRNAs be administered clinically?  Presumably by transtympanic injection.

Toxicology studies would be required to determine the safety of microbubble-siRNAs.

Round 2

Reviewer 2 Report

Comments and Suggestions for Authors

One minor revision should specify in the legend for Figure 9 that the area of the cochlea shown is the middle turn. 
